# Towards Seamless Adaptation of Pre-trained Models for Visual Place Recognition

**Feng Lu**[1,2]**, Lijun Zhang**[3]**, Xiangyuan Lan**[2*]**, Shuting Dong**[1]**, Yaowei Wang**[2]**, Chun Yuan**[1*]

[1]Tsinghua Shenzhen International Graduate School, Tsinghua University
[2]Peng Cheng Laboratory  [3]University of Chinese Academy of Sciences
{lf22@mails,yuanc@sz}.tsinghua.edu.cn   lanxy@pcl.ac.cn

## Abstract

Recent studies show that vision models pre-trained in generic visual learning tasks with large-scale data can provide useful feature representations for a wide range of visual perception problems. However, few attempts have been made to exploit pre-trained foundation models in visual place recognition (VPR). Due to the inherent difference in training objectives and data between the tasks of model pre-training and VPR, how to bridge the gap and fully unleash the capability of pre-trained models for VPR is still a key issue to address. To this end, we propose a novel method to realize seamless adaptation of pre-trained models for VPR. Specifically, to obtain both global and local features that focus on salient landmarks for discriminating places, we design a hybrid adaptation method to achieve both global and local adaptation efficiently, in which only lightweight adapters are tuned without adjusting the pre-trained model. Besides, to guide effective adaptation, we propose a mutual nearest neighbor local feature loss, which ensures proper dense local features are produced for local matching and avoids time-consuming spatial verification in re-ranking. Experimental results show that our method outperforms the state-of-the-art methods with less training data and training time, and uses about only 3% retrieval runtime of the two-stage VPR methods with RANSAC-based spatial verification. It ranks 1st on the MSLS challenge leaderboard (at the time of submission). The code is released at https://github.com/Lu-Feng/SelaVPR.

## 1 Introduction

Visual place recognition (VPR), also known as image localization (Liu et al., 2019) or visual geo-localization (Berton et al., 2022b), aims at coarsely estimating the location of a query place image by searching for its best match from a database of geo-tagged images. VPR has long been studied in robotics and computer vision communities, motivated by its wide applications in mobile robot localization (Xu et al., 2020) and augmented reality (Middelberg et al., 2014), etc. The main challenges of the VPR task include condition (e.g., illumination and weather) changes, viewpoint changes, and perceptual aliasing (Lowry et al., 2015) (hard to differentiate similar images from different places).

The VPR task is typically addressed by using image retrieval and matching approaches (Arandjelovic et al., 2016; Cao et al., 2020) with global or/and local descriptors to represent images. The aggregation algorithms like VLAD (Jégou et al., 2010; Lowry & Andreasson, 2018; Khaliq et al., 2019) are usually used to aggregate/pool local features into a vector as the global feature. Such compact global features facilitate fast place retrieval and are robust against viewpoint variations. However, these global features neglect spatial information, making VPR methods based on them prone to perceptual aliasing. A promising solution (Cao et al., 2020; Hausler et al., 2021; Wang et al., 2022a), i.e., two-stage VPR, is to retrieve top-k candidate results in the database using global features, then re-rank these candidates by matching local features. Moreover, VPR model training follows the "pre-training then finetuning" paradigm. Most VPR models are initialized using model parameters pre-trained on ImageNet (Deng et al., 2009) and fine-tuned on the VPR datasets, such as MSLS (Warburg et al., 2020). As models and training datasets continue to expand, the training becomes more costly in both computation and memory footprint.

---

*Corresponding authors.

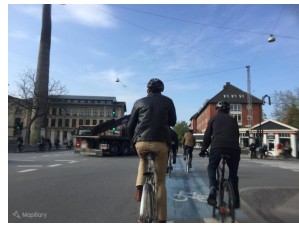 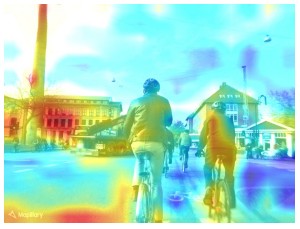 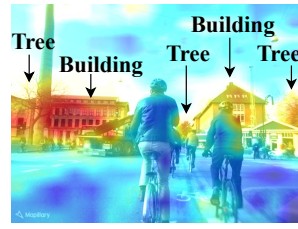

(a) Input image          (b) Result of pre-trained model          (c) Result of our method

Figure 1: Attention map visualizations of the pre-trained foundation model (DINOv2) and our model. The pre-trained model pays attention to some regions (e.g. dynamic riders) that are useless to identify places. Our method focuses on discriminative regions (buildings and trees).

Recently, foundation models (Radford et al., 2021; Yuan et al., 2021; Oquab et al., 2023) achieved remarkable performance on many computer vision tasks given their ability to produce well-generalized representations. However, the image representation produced by the pre-trained model is susceptible to useless (even harmful) dynamic objects (e.g. pedestrians and vehicles), and tends to ignore some static discriminative backgrounds (e.g. buildings and vegetation), as shown in Fig. 1 (b). A robust VPR model should focus on the static discriminative landmarks (Chen et al., 2017b) rather than the dynamic foreground. This results in a gap between pre-training and VPR tasks. Meanwhile, full fine-tuning the foundation model on downstream datasets might forget previously learned knowledge and damage the excellent transferability, i.e., catastrophic forgetting. An effective method to address this issue is parameter-efficient transfer learning (Houlsby et al., 2019; Lester et al., 2021), which has not been studied in the VPR area. Besides, most foundation models do not directly produce (dense) local features, which is typically required in the re-ranking of two-stage VPR methods.

In this paper, we propose a novel method to realize **Se**amless **a**daptation of pre-trained foundation models for the VPR task, named **SelaVPR**. By adding a few tunable lightweight adapters to the frozen pre-trained model, we achieve an efficient hybrid global-local adaptation to get both global features for retrieving candidate places and local features for re-ranking. Specifically, the global adaptation is achieved by adding adapters after the multi-head attention layer and in parallel to the MLP layer in each transformer block. The local adaptation is implemented by adding up-convolutional layers after the entire transformer backbone to upsample the feature map. Additionally, we propose a mutual nearest neighbor local feature loss, which can be combined with the commonly used triplet loss to optimize the network. The proposed SelaVPR feature representation focuses on the discriminative landmarks, which is critical to identifying places. Furthermore, we can directly match the local features without spatial verification, making the re-ranking much faster than mainstream two-stage VPR methods. Our main **contributions** are highlighted as follows:

**1)** We propose a hybrid global-local adaptation method to seamlessly adapt pre-trained foundation models to produce both global and local features for the VPR task. The proposed SelaVPR feature representation can focus on discriminative landmarks and ignore the regions irrelevant to distinguishing places, thus closing the gap between the pre-training and VPR tasks.

**2)** We also propose a mutual nearest neighbor local feature loss to train the local adaptation module, which is combined with global feature loss for fine-tuning. The obtained local features can be directly used in cross-matching for re-ranking, without time-consuming geometric verification.

**3)** Our method outperforms state-of-the-art methods on several VPR benchmarks (ranks 1st on MSLS challenge leaderboard) using less training data and training time. And it only consumes 3% retrieval runtime of the mainstream two-stage methods with RANSAC-based geometric verification.

## 2   RELATED WORK

**Visual Place Recognition:** The traditional VPR approaches perform nearest neighbor search using global features to find the most similar place. The global features are commonly produced using aggregation algorithms, such as Bag of Words (Angeli et al., 2008) and VLAD (Jégou et al., 2010), to process the hand-crafted features like SURF (Bay et al., 2008). With the advancement of deep learning techniques, many works (Sünderhauf et al., 2015; Jin Kim et al., 2017; Chen et al., 2017a;b;

Naseer et al., 2017; Garg et al., 2017; 2018; Xin et al., 2019; Yin et al., 2019; Lu et al., 2021; Leyva-Vallina et al., 2021; 2023; Ali-Bey et al., 2023) have employed a variety of deep features for the VPR task. Some works integrated the aggregation methods into neural networks (Arandjelovic et al., 2016; Yu et al., 2019; Peng et al., 2021), and improved training strategies (Ge et al., 2020; Berton et al., 2022a; Ali-bey et al., 2022), to achieve better performance. Nevertheless, most one-stage (i.e. global retrieval) VPR approaches are prone to perceptual aliasing due to the use of aggregated features while neglecting spatial information. One recent work (Keetha et al., 2023) also used pre-trained foundation models for the VPR task. However, this work did not perform any fine-tuning, making it difficult to fully unleash the capability of these models for VPR.

Recently, the two-stage (i.e. hierarchical) VPR methods (Hausler & Milford, 2020; Garg & Milford, 2021; Hausler et al., 2021; Berton et al., 2021; Shen et al., 2022; Wang et al., 2022a; Lu et al., 2023; Shen et al., 2023; Zhu et al., 2023) have become popular. These approaches typically retrieved top-k candidate images over the whole database using compact global feature representation, such as NetVLAD (Arandjelovic et al., 2016) or Generalized Mean (GeM) pooling (Radenović et al., 2018), then re-ranked candidates by performing local matching between the query image and each candidate using local descriptors. However, most of these methods required geometric consistency verification after local matching (Cao et al., 2020; Hausler et al., 2021; Wang et al., 2022a) or taking into account spatial constraints during matching (Berton et al., 2021; Lu et al., 2023; Zhu et al., 2023), which greatly increases the runtime burden. The recent visual foundation model (Oquab et al., 2023) has shown an excellent ability to match similar semantic patch-level features across domains. In this work, we attempt to match local features produced by the foundation model for re-ranking without spatial verification, thereby significantly speeding up the retrieval process in VPR.

**Parameter-efficient Transfer Learning:** Recent work (Radford et al., 2021; Caron et al., 2021; Wang et al., 2022b; Oquab et al., 2023) demonstrated the visual foundation model can produce powerful feature representation and achieve excellent performance on multiple tasks. These works commonly trained the ViT (Dosovitskiy et al., 2020) model or its variants with large quantities of parameters on huge amounts of data. The parameter-efficient transfer learning (PETL) (Houlsby et al., 2019), first proposed in natural language processing, is an effective way to adapt foundation models to various downstream tasks, which can reduce training computation costs and avoid catastrophic forgetting. The main PETL methods fall broadly into three categories: adding task-specific adapters (Houlsby et al., 2019), prompt tuning (Lester et al., 2021), and Low-Rank Adaptation (LoRA) (Hu et al., 2021). We follow the first in this work. Although several adapter-based approaches (Jie & Deng, 2022; Chen et al., 2022; Pan et al., 2022; Yang et al., 2023; Khan & Fu, 2023; Xu et al., 2023; Park et al., 2023) have been proposed to perform various computer vision tasks, to the best of our knowledge, this work is among the first to use the hybrid global-local adaptation to produce both global features and local features, and apply them to address the challenges in VPR.

## 3 PROPOSED METHOD

This section describes the proposed SelaVPR for two-stage VPR. We first introduce ViT and its use to produce place image representation. Then, we propose the global adaptation, local adaptation, and local matching re-ranking to achieve two-stage VPR. Finally, we present the loss for fine-tuning.

### 3.1 PRELIMINARY

The Vision Transformer (ViT) and its variants have proven to be powerful for a variety of computer vision tasks including VPR. In this work, we adapt the ViT-based pre-trained foundation model DINOv2 (Oquab et al., 2023) for VPR, so here we give a brief overview of ViT.

Given an input image, ViT first slices it into $N$ patches and linearly projects them to $D$-dim patch embeddings $x_p \in \mathcal{R}^{N \times D}$, then prepends a learnable [class] token to $x_p$ as $x_0 = [x_{class}; x_p] \in \mathcal{R}^{(N+1) \times D}$. After adding positional embeddings to preserve the positional information, $x_0$ is fed into a series of transformer blocks to produce the feature representation. A standard transformer block mainly includes multi-head attention (MHA), multi-layer perceptron ( MLP), and LayerNormalization (LN) layers, as shown in Fig. 2 (a). For the input token sequence, its change process passing through a transformer block is: The MHA is first applied to compute attentional features,

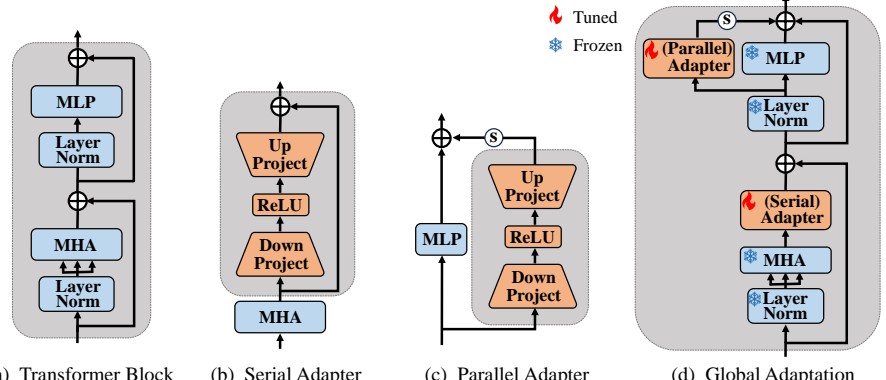

Figure 2: Illustration of the global adaptation. We add a serial adapter (b) after the MHA layer and a parallel adapter (c) in parallel to the MLP layer in each standard transformer block (a) to achieve global adaptation.

then MLP is utilized to realize the feature nonlinearization and dimension transformation. It is formulated as:

$$x'_l = \text{MHA}(\text{LN}(x_{l-1})) + x_{l-1} \tag{1}$$

$$x_l = \text{MLP}(\text{LN}(x'_l)) + x'_l, \tag{2}$$

where $x_{l-1}$ and $x_l$ are the output of the $(l-1)$-th and $l$-th transformer block.

For the feature map output by CNN models, a common practice of the two-stage VPR method is to use NetVLAD or GeM to aggregate it into a global feature for candidate retrieval and also treat it as dense local (patch) features for re-ranking. For the ViT model, the output consists of one class token and $N$ patch tokens, where the class token can be directly used as the global feature to represent places. Meanwhile, $N$ patch tokens can also be reshaped as a feature map (similar to CNN) to restore spatial position. In this work, instead of using the class token as the global feature, we use GeM to pool the feature map into the global feature. The reason is explained in Appendix C.

## 3.2 GLOBAL ADAPTATION

Although pre-trained foundation models are capable of powerful feature representation, direct use of them in VPR cannot fully unleash their capability due to the gap between the pre-training and VPR tasks. To address it, we introduce the global adaptation to adapt the pre-trained model so that the feature representation can focus on the static discriminative regions that are beneficial to VPR.

Inspired by previous adapter-based parameter-efficient fine-tuning works (Houlsby et al., 2019; Chen et al., 2022; Yang et al., 2023), we design our global adaptation as shown in Fig. 2 (d). Specifically, we add two adapters in each transformer block. Each adapter is a bottleneck module, which first uses the fully connected layer to down-project the input to a smaller dimension, then applies a ReLU activation and up-projects it back to the original dimension. The first adapter is a serial adapter that is added after the MHA layer and has a skip-connection internally. The second adapter is a parallel adapter that is connected in parallel to the MLP layer multiplied by a scaling factor $s$. The computation of each global adapted transformer block can be denoted as

$$x'_l = \text{Adapter1}(\text{MHA}(\text{LN}(x_{l-1}))) + x_{l-1} \tag{3}$$

$$x_l = \text{MLP}(\text{LN}(x'_l)) + s \cdot \text{Adapter2}(\text{LN}(x'_l)) + x'_l. \tag{4}$$

The output of the last transformer block is fed into an LN layer as the final output of the entire global adapted ViT backbone. We discard the class token and reshape patch tokens as the produced feature map $fm$. The L2-normalized GeM global feature used to retrieve candidates can be written as

$$\boldsymbol{f}^g = \text{L2}(\text{GeM}(fm)). \tag{5}$$

The global adapted foundation model can produce feature representations that focus on discriminative landmarks and ignore dynamic interference. This bridges the gap between the model pre-training and VPR tasks, and greatly boosts the performance of foundation models in the VPR task.

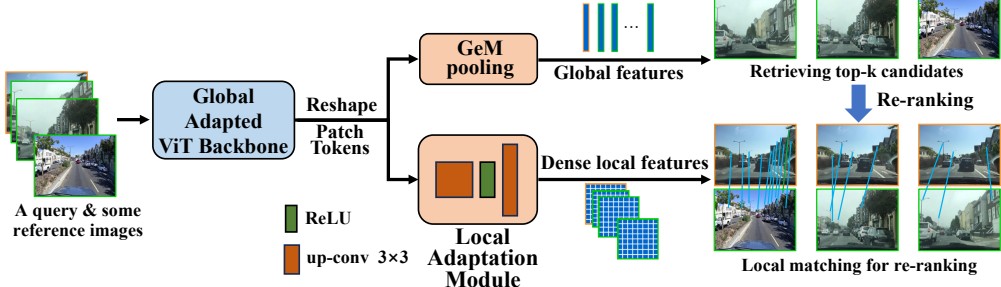

Figure 3: Illustration of the local adaptation and our two-stage VPR pipeline. The global adapted ViT backbone is applied to extract the feature map. We first use GeM to pool the feature map into the global feature for candidate retrieval. The local adaptation module after the backbone is achieved using up-conv layers, which upsample the feature map to yield dense local features. Then we cross-match the local features between the query image and each candidate for re-ranking.

### 3.3 LOCAL ADAPTATION

Two-stage VPR methods typically match dense local features to re-rank candidate places for boosting performance. In the above introduction, we have obtained the feature map output by the global adapted ViT backbone, which can also be regarded as coarse-grained patch-level features. However, in order to achieve promising performance improvements through (local matching) re-ranking, more fine-grained dense local features are required. To achieve this, we propose the local adaptation, which is achieved by an up-sampling module after the ViT backbone, as shown in Fig. 3. To be specific, this module consists of two up-convolutional (up-conv) layers and a ReLU layer in the middle. The height and width of the feature map will approximately double after passing through each up-conv layer, while the channel dimension will be reduced. Finally, for the input image of size 224×224 pixels, this module adjusts the 16×16×1024-dim feature map output by ViT-L/14 backbone to 61×61×128-dim, and performs L2 normalization in the channel dimension (intraL2) to yield dense local features, i.e., a dense 61×61 grid of 128-dim local features $\boldsymbol{f}^l$. Formally, it can be represented as

$$\boldsymbol{f}^l = \text{LocalAdaptation}(fm) = \text{intraL2}(\text{up-conv2}(\text{ReLU}(\text{up-conv1}(fm)))). \tag{6}$$

### 3.4 LOCAL MATCHING FOR RE-RANKING

Our two-stage place retrieval pipeline is shown in Fig. 3. After obtaining the global features and local features, we first compute L2 distance to perform the similarity search in the global feature space over the database to get the top-k most similar candidate images. For the local features matching between the query image $q$ and a candidate image $c$, we search for mutual nearest neighbor matches by cross-matching. Since local features are L2 normalized, the inner product equivalent to cosine similarity is used to measure local feature similarity. That is

$$\boldsymbol{s}_{qc}(i,j) = \boldsymbol{f}_q^l(i) \cdot \boldsymbol{f}_c^l(j) \quad i,j \in \{1,2,...,N'\} \quad (N' = 61 \times 61), \tag{7}$$

where $\boldsymbol{f}_q^l(i)$ is the $i$-th local feature in query image $q$, $\boldsymbol{f}_c^l(j)$ is the $j$-th local feature in candidate image $c$, and $\boldsymbol{s}_{qc}(i,j)$ is the local feature similarity between them. The mutual nearest neighbor matches set $\mathcal{M}$ is defined as

$$\mathcal{M} = \{(u,v): \ u = \arg\max_i \boldsymbol{s}_{qc}(i,v), \ v = \arg\max_j \boldsymbol{s}_{qc}(u,j)\}. \tag{8}$$

That is, the $u$-th feature in image $q$ and the $v$-th feature in image $c$ are the best matches for each other. The obtained nearest neighbor matches set in previous VPR work (Hausler et al., 2021; Wang et al., 2022a) exists a large number of false matches. So they apply geometric verification (e.g. RANSAC (Fischler & Bolles, 1981)) to remove outliers (i.e. false matches) and use the number of inliers as the image similarity score, which is time-consuming. Due to the powerful representation ability of the foundation model and the superiority of the ViT in capturing long-distance feature dependencies, the number of false matches in this work is actually not enough to affect the performance of re-ranking. So we directly use the number of matches (i.e., $|\mathcal{M}|$) as the image similarity score for re-ranking.

### 3.5 Loss

For the loss designed to optimize the model to produce global features (denoted as global loss $L_g$), we follow the triplet loss used in the previous works (Arandjelovic et al., 2016; Wang et al., 2022a):

$$L_g = \sum_j l(\|\boldsymbol{f}_q^g - \boldsymbol{f}_p^g\| + m - \|\boldsymbol{f}_q^g - \boldsymbol{f}_{n_j}^g\|), \qquad (9)$$

where $l(x) = \max(x, 0)$, i.e. hinge loss. $m$ is the margin. $\boldsymbol{f}_q^g$, $\boldsymbol{f}_p^g$, and $\boldsymbol{f}_{n_j}^g$ are the global features of query, positive, and hard negative samples, respectively.

However, what we use to measure image similarity in local matching re-ranking is the number of mutual matches, which is a discrete integer. It is intractable to directly optimize a loss function of discrete integer variables within a deep learning model because of its non-differentiability. So we compromise to optimize the network so that the resulting mutual matching local features are more similar, and design a mutual nearest neighbor local feature loss $L_l$ as

$$L_l = \sum_j l(-\frac{\sum_{(u,v)\in\mathcal{M}} \boldsymbol{s}_{qp}(u, v)}{|\mathcal{M}|} + \frac{\sum_{(u',v')\in\mathcal{M}'} \boldsymbol{s}_{qn_j}(u', v')}{|\mathcal{M}'|}). \qquad (10)$$

This local loss maximizes the average local feature similarity in the mutual matches set $\mathcal{M}$ of the query image and positive image, and minimizes that in the matches set $\mathcal{M}'$ of the query and negative images. It makes the produced local features more suitable for local matching. We obtain the final loss $L$ through combining the global loss $L_g$ and local loss $L_l$ by weight $\lambda$ as

$$L = L_g + \lambda L_l. \qquad (11)$$

## 4 Experiments

### 4.1 Datasets and Performance Evaluation

Several VPR benchmark datasets mainly including Tokyo24/7, MSLS, and Pitts30k are used in our experiments. Table 1 summarizes their main information. **Tokyo24/7** (Torii et al., 2015) includes about 76k database images and 315 query images captured from urban scenes with drastic illumination changes. **Mapillary Street-Level Sequences (MSLS)** (Warburg et al., 2020) consists of more than 1.6 million images collected in urban, suburban and natural scenes over 7 years. We assess

Table 1: Summary of the main evaluation datasets.

| Dataset | Description | Number | |
|---|---|---|---|
| | | Database | Queries |
| Tokyo24/7 | urban, day/night | 76k | 315 |
| MSLS-val | urban, suburban | 19k | 740 |
| MSLS-challenge | long-term | 39k | 27092 |
| Pitts30k-test | urban, panorama | 10k | 6816 |

models on both MSLS-val and MSLS-challenge (an online test set without released labels) sets. **Pittsburgh (Pitts30k)** (Torii et al., 2013) contains 30k reference images and 24k query images in the train, val and test sets, and exhibits severe viewpoint changes. More details are in Appendix J.

We evaluate the recognition performance using Recall@N (R@N), which is the percentage of queries for which at least one of the N retrieved images is the right result. The threshold is set to 25 meters and 40° for MSLS, 25 meters for Tokyo24/7 and Pitts30k, following standard evaluation procedure (Warburg et al., 2020; Arandjelovic et al., 2016).

### 4.2 Implementation Details

We use the DINOv2 based on ViT-L/14 as the foundation model and conduct all experiments on an NVIDIA GeForce RTX 3090 GPU using PyTorch. Fed a 224×224 image, the model produces a 1024-dim global feature and a dense grid of 128-dim local features. The bottleneck ratio of the adapters in ViT blocks is 0.5 and the scaling factor $s$ in Eq. 4 is set to 0.2. We use 3×3 up-conv with stride=2 and padding=1 in the local adaptation module. The output channels of the first and second up-conv layers are 256 and 128, respectively. Following other two-stage methods, we re-rank the top-100 candidates to yield final results. We train our models using the Adam optimizer with the learning rate set as 0.00001 and batch size set as 4. When the R@5 on the validation set does not have improvement within 3 epochs, the training is terminated. For MSLS, we set an epoch

Table 2: Comparison to state-of-the-art methods on benchmark datasets. The best is highlighted in **bold** and the second is underlined.

| Method | Tokyo24/7 | | | MSLS-val | | | MSLS-challenge | | | Pitts30k-test | | |
|---|---|---|---|---|---|---|---|---|---|---|---|---|
| | R@1 | R@5 | R@10 | R@1 | R@5 | R@10 | R@1 | R@5 | R@10 | R@1 | R@5 | R@10 |
| NetVLAD | 60.6 | 68.9 | 74.6 | 53.1 | 66.5 | 71.1 | 35.1 | 47.4 | 51.7 | 81.9 | 91.2 | 93.7 |
| SFRS | 81.0 | 88.3 | 92.4 | 69.2 | 80.3 | 83.1 | 41.6 | 52.0 | 56.3 | 89.4 | 94.7 | 95.9 |
| CosPlace | 81.9 | 90.2 | 92.7 | 82.8 | 89.7 | 92.0 | 61.4 | 72.0 | 76.6 | 88.4 | 94.5 | 95.7 |
| MixVPR | 85.1 | 91.7 | 94.3 | 88.0 | 92.7 | 94.6 | 64.0 | 75.9 | 80.6 | 91.5 | 95.5 | 96.3 |
| SelaVPR(global) | 81.9 | 94.9 | 96.5 | 87.7 | 95.8 | 96.6 | 69.6 | 86.9 | 90.1 | 90.2 | 96.1 | 97.1 |
| SP-SuperGlue | 88.2 | 90.2 | 90.2 | 78.1 | 81.9 | 84.3 | 50.6 | 56.9 | 58.3 | 87.2 | 94.8 | 96.4 |
| Patch-NetVLAD-s | 78.1 | 83.8 | 87.0 | 77.8 | 84.3 | 86.5 | 48.1 | 59.4 | 62.3 | 87.5 | 94.5 | 96.0 |
| Patch-NetVLAD-p | 86.0 | 88.6 | 90.5 | 79.5 | 86.2 | 87.7 | 48.1 | 57.6 | 60.5 | 88.7 | 94.5 | 95.9 |
| TransVPR | 79.0 | 82.2 | 85.1 | 86.8 | 91.2 | 92.4 | 63.9 | 74.0 | 77.5 | 89.0 | 94.9 | 96.2 |
| StructVPR | - | - | - | 88.4 | 94.3 | 95.0 | 69.4 | 81.5 | 85.6 | 90.3 | 96.0 | 97.3 |
| $R^2$Former | 88.6 | 91.4 | 91.7 | 89.7 | 95.0 | 96.2 | 73.0 | 85.9 | 88.8 | 91.1 | 95.2 | 96.3 |
| SelaVPR (ours) | **94.0** | **96.8** | **97.5** | **90.8** | **96.4** | **97.2** | **73.5** | **87.5** | **90.6** | **92.8** | **96.8** | **97.7** |

as passing 30k queries, whereas Pitts30k is passing 5k queries. In model training, we define the potential positive images as the reference images that are within 10 meters from the query image, while the definite negative images are those further than 25 meters. Two hard negative images from 1000 randomly chosen definite negatives are used in the triplet loss. We empirically set the margin $m = 0.1$ in Eq. 9, the weight $\lambda = 1$ in Eq. 11. For the experiments in Section 4.3 (i.e. comparisons with other methods), we fine-tune our model on MSLS-train to test on MSLS-val/challenge, and further fine-tune it on Pitts30k-train to test on Pitts30k-test and Tokyo24/7 (as in $R^2$Former (Zhu et al., 2023)). For the ablation experiments in other sections, the models tested on Pitts30k-test and Tokyo24/7 are directly fine-tuned on Pitts30k-train by default.

## 4.3 COMPARISONS WITH STATE-OF-THE-ART METHODS

In this section, we compare the proposed SelaVPR method with several state-of-the-art (SOTA) VPR methods, including four one-stage methods using global feature retrieval: NetVLAD (Arandjelovic et al., 2016), SFRS (Ge et al., 2020), CosPlace (Berton et al., 2022a) and MixVPR (Ali-Bey et al., 2023), as well as five two-stage methods with re-ranking: SP-SuperGlue (DeTone et al., 2018; Sarlin et al., 2020), Patch-NetVLAD (Hausler et al., 2021), TransVPR (Wang et al., 2022a), StructVPR (Shen et al., 2023) and $R^2$Former (Zhu et al., 2023). The details of these methods are in Appendix K. Note that CosPlace and MixVPR are trained on individually constructed large-scale datasets. TransVPR and $R^2$Former are transformer-based methods. $R^2$Former ranked first in the leaderboard of the MSLS place recognition challenge before our method was proposed. We also show the global retrieval result without re-ranking using the proposed SelaVPR, and denote it as SelaVPR(global). The quantitative results are shown in Table 2. Since the code for StructVPR has not yet been released, we use the results reported in the paper (Shen et al., 2023). The proposed method achieves the best R@1/R@5/R@10 on all datasets.

When using only global features for direct retrieval, SelaVPR(global) significantly outperforms other one-stage methods on all datasets for R@5 and R@10, including SFRS using complex training strategies and CosPlace/MixVPR trained on purpose-built large-scale datasets. SelaVPR(global) also outperforms all other two-stage methods on Tokyo24/7, MSLS-val, and MSLS-challenge for R@5 and R@10. This fully demonstrates that adapting the foundation model can provide powerful feature representation, which is a novel way to achieve SOTA one-stage VPR. Although SelaVPR(global) does not achieve very good R@1 on Tokyo24/7 where the lighting changes drastically, the complete SelaVPR method outperforms other methods by a large margin after local feature re-ranking. This illustrates the necessity of using local matching (local adaptation) for VPR in extreme environments. The complete SelaVPR method significantly outperforms SOTA methods on Tokyo24/7 and Pitts30k with absolute R@1 improvement of 5.4% and 1.3% respectively. Meanwhile, it also ranks 1st on the leaderboard of MSLS place recognition challenge (see Appendix L). Comparisons to SOTA methods on more datasets are in Appendix F (SelaVPR achieves near-perfect results on St. Lucia). Fig. 4 qualitatively demonstrates that our approach is highly robust in challenging scenes. Benefiting from the visual foundation model and sensible adaptation, SelaVPR achieves absolute performance advantages on a variety of datasets without complex training strategies or purpose-built large-scale training datasets.

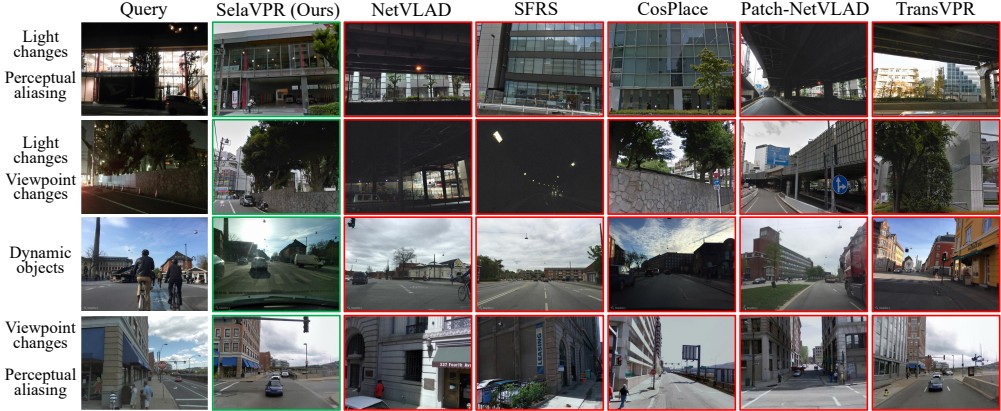

Figure 4: Qualitative results. In these challenging examples (containing condition changes, viewpoint changes, dynamic objects, etc.), the proposed SelaVPR successfully returns the right database images, while all other methods produce incorrect results.

Table 3: The single query runtime comparison of two-stage methods on Pitts30k-test.

| Method | Extraction Time (s) | Matching Time (s) | Total Time (s) |
|---|---|---|---|
| SP-SuperGlue | 0.042 | 6.639 | 6.681 |
| Patch-NetVLAD-s | 0.186 | 0.551 | 0.737 |
| Patch-NetVLAD-p | 0.412 | 10.732 | 11.144 |
| TransVPR | **0.008** | 3.010 | 3.018 |
| SelaVPR (ours) | 0.027 | **0.085** | **0.112** |

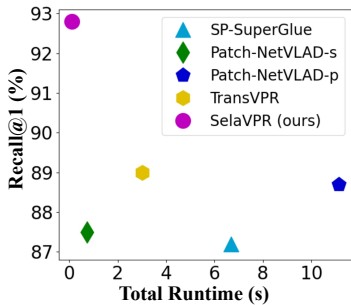

Figure 5: R@1-runtime comparison.

Efficiency is another metric for evaluating the VPR methods. In Table 3, we compare the runtime (feature extraction time and matching/retrieval time) of our method with other two-stage methods on Pitts30k-test. Patch-NetVLAD-p and TransVPR are representative two-stage methods using RANSAC for spatial verification in re-ranking. SP-SuperGlue uses neural networks to match local features. TransVPR is fast at extracting features, while SelaVPR is slower (but faster than other methods) due to the use of the ViT/L backbone. However, for matching/retrieval, since our method does not require time-consuming spatial verification in re-ranking, its runtime is less than 3% of TransVPR and only about 1% of SP-SuperGlue and Patch-NetVLAD-p. The total runtime of our method is less than 4% of TransVPR. Although Patch-NetVLAD-s uses a Rapid Spatial Scoring method for fast verification, it is still significantly slower than ours. Fig. 5 simultaneously shows the total runtime and R@1 on Pitts30k. Due to the absolute advantages in both performance and efficiency, our SelaVPR method is able to pave the way for real-world large-scale VPR applications.

## 4.4 ABLATION STUDY

In this section, we perform a series of ablation experiments to demonstrate the necessity of fine-tuning and verify the effectiveness of the proposed global adaptation and local adaptation:

- **DINOv2-GeM:** Using the pre-trained DINOv2 backbone (freeze parameter) and GeM pooling, i.e., baseline.
- **Tuned-DINOv2-GeM:** full fine-tuned DINOv2-GeM.
- **Global-Adaptation:** Global adapted DINOv2-GeM with our global adapation.
- **Local-Adaptation:** Using the DINOv2-GeM (freeze parameter) to retrieve candidates, and adding the local adaptation module after it to produce local features for re-ranking.
- **SelaVPR:** The complete (hybrid global-local adaptation) method.

The results are shown in Table 4. Note that we directly use the model fine-tuned on Pitts30k-train to test on Pitts30k and Tokyo24/7 in this section. The results are slightly lower than that in Table 2 (but still SOTA).

Table 4: Comparison of different ablated versions.

| Ablated version | Pitts30k-test | | | Tokyo24/7 | | | MSLS-val | | |
|---|---|---|---|---|---|---|---|---|---|
| | R@1 | R@5 | R@10 | R@1 | R@5 | R@10 | R@1 | R@5 | R@10 |
| DINOv2-GeM | 81.3 | 91.0 | 93.8 | 67.3 | 85.1 | 89.8 | 44.7 | 55.9 | 59.3 |
| Tuned-DINOv2-GeM | 85.3 | 92.7 | 94.7 | 65.7 | 78.1 | 83.8 | 79.7 | 90.3 | 92.2 |
| Global-Adaptation | 87.3 | 94.6 | 96.6 | 77.8 | 87.6 | 91.7 | 87.4 | 95.9 | 96.9 |
| Local-Adaptation | 87.2 | 93.9 | 96.1 | 87.6 | 94.6 | **95.9** | 67.2 | 75.0 | 76.6 |
| SelaVPR | **91.4** | **96.5** | **97.8** | **93.3** | **95.2** | 95.6 | **90.8** | **96.4** | **97.2** |

The pre-trained DINOv2-GeM achieves decent results on Pitts30k, which has few dynamic objects on the place image. The performance is improved after full fine-tuning (i.e. Tuned-DINOv2-GeM), indicating that fine-tuning is necessary to make the produced feature representation more suitable for the VPR task even without dynamic interference. However, Tuned-DINOv2-GeM performs worse than DINOv2-GeM on Tokyo24/7. It is because there is a generalization gap between Tokyo24/7 and Pitts30k-train, and full fine-tuning damages the excellent transferability of the pre-trained foundation model. Compared with Tuned-DINOv2-GeM, Global-Adaptation only tunes a few newly added adapters (with backbone frozen), which reduces the training consumption and always improves performance (retaining the generalization ability of the foundation model). Global-Adaptation achieves absolute R@1 improvement of 10.5% on Tokyo24/7. Besides, Local-Adaptation only adds a local adaptation module (with the proposed local loss for training) after the frozen DINOv2 to produce local features for re-ranking. It can also obviously improve performance, especially on Tokyo24/7 (20.3% absolute R@1 improvement), which shows day-night changes. The complete SelaVPR method combines the advantages of Glocal-Adaptation and Local-Adaptation, and achieves absolute R@1 improvement of 10.1% on Pitts30k and 26.0% on Tokyo24/7.

MSLS is a dataset with many dynamic objects (e.g. vehicles and pedestrians), on which the pre-trained DINOv2-GeM gets terrible results. By adding a local adaptation module after DINOv2 (Local-Adaptation) to get dense local features for re-ranking, we can get 22.5% absolute performance improvement for R@1, which is still not ideal. Using full fine-tuning can even bring significant performance improvement (35.0% for R@1), and the improvement yielded by Global-Adaptation is more obvious (42.7% for R@1). More importantly, the complete SelaVPR method achieves 2× higher R@1.

Overall, full fine-tuning the pre-trained foundation model on these datasets usually results in better performance (unless there is a domain gap between the training and test set), indicating that the first problem should be solved when applying a foundation model to the VPR task is to make the output feature representation suitable for VPR (focusing on regions that help differentiate places), i.e. bridge the gap between pre-training and VPR tasks. The second is the catastrophic forgetting that will be encountered in fine-tuning the pre-trained models, which can be addressed by parameter-efficient fine-tuning instead of full fine-tuning. Additionally, the local matching for re-ranking is also necessary, especially on datasets that show drastic condition variations (e.g. Tokyo24/7). The proposed global adaptation and local adaptation can work together to solve these problems well.

More experiments and analyses are in the Appendix, e.g., the comparisons of tunable parameters, data efficiency, and training time.

## 5 CONCLUSIONS

In this paper, we introduced a novel hybrid adaptation method to seamlessly adapt pre-trained foundation models for the VPR task, which is composed of the global adaptation and local adaptation. The feature representation produced by the adapted foundation model is more focused on discriminative landmarks to differentiate places, thus bridging the gap between the pre-training and VPR tasks. The local adaptation enables the model to output proper dense local features, which can be used for local matching re-ranking in two-stage VPR to greatly boost performance. The experimental results demonstrated that the proposed SelaVPR method outperforms previous SOTA methods on VPR benchmark datasets (with a large margin on Tokyo24/7 and Pitts30k). Meanwhile, since our approach eliminates the reliance on time-consuming spatial validation in re-ranking, it costs only 3% retrieval time of RANSAC-based two-stage methods. We believe that the proposed SelaVPR method provides a promising way to address the VPR task in real-world large-scale applications.

ACKNOWLEDGMENTS

This work was supported by the National Key R&D Program of China (2022YFB4701400/4701402), SSTIC Grant (KJZD20230923115106012), Shenzhen Key Laboratory (ZDSYS20210623092001004), Beijing Key Lab of Networked Multimedia, and the Project of Peng Cheng Laboratory (PCL2023A08).

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

Table 5: Tunable parameters, training epoch, and training time of different methods on Pitts30k.

| Method | Tunable Param (M) | epoch | time (h) | R@1 | R@5 |
|---|---|---|---|---|---|
| ResNet50-GeM | **7** | 9 | 1.63 | 82.3 | 91.9 |
| Tuned-DINOv2-GeM | 304 | 2 | 0.62 | 85.3 | 92.7 |
| SelaVPR | 53 | 1 | 0.30 | **91.4** | **96.5** |
| SelaVPR* | 53 | **0.4** | **0.12** | 91.0 | 96.2 |

Table 6: The results of using different adapters in the global adaptation.

| Method | Pitts30k-test | | | MSLS-val | | |
|---|---|---|---|---|---|---|
| | R@1 | R@5 | R@10 | R@1 | R@5 | R@10 |
| only serial adapter | 90.9 | 96.2 | 97.4 | **90.9** | 95.8 | 96.9 |
| only parallel adapter | 91.3 | **96.6** | 97.7 | 89.7 | 96.2 | 96.6 |
| both adapters | **91.4** | 96.5 | **97.8** | 90.8 | **96.4** | **97.2** |

# A    COMPARISONS OF TUNABLE PARAMETERS, DATA EFFICIENCY, AND TRAINING TIME

In this section, we evaluate the tunable parameters, data efficiency, and training time of our model. We use the ResNet50 with a GeM pooling in the benchmark implementation (Berton et al., 2022b) (ResNet50-GeM) and the full fine-tuning DINOv2-GeM (Tuned-DINOv2-GeM) as reference. This experiment is conducted on Pitts30k and the results are shown in Table 5. The tunable parameters of our SelaVPR are only about 1/6 of full fine-tuning DINOv2 (Tuned-DINOv2-GeM). Although our model has more tunable parameters than the ResNet50-GeM (CNN backbone), we require significantly less training data and time, and achieve a 9.1% higher R@1. Note that one epoch only contains a part of the training set (5000 triples). To further study the data efficiency of our model, we train it with only 2000 triples (i.e., 0.4 epoch) and denote it as SelaVPR*. It still achieves SOTA performance.

# B    ADDITIONAL ABLATION EXPERIMENTS FOR THE GLOBAL ADAPTATION

In this section, we further conduct ablation experiments to study the effect of reducing the number of tunable parameters in the global adaptation. Table 6 shows the results of using different adapters, i.e. only serial adapter, only parallel adapter, or both adapters. Using both adapters achieves the best results overall, but we can still get good performance with just either adapter. Table 7 shows the results of setting different bottleneck ratios in each adapter. Setting the ratio to 0.5 gets the best results overall, but we can still achieve good performance with a 0.25 bottleneck ratio. These experimental findings show that promising results can be obtained even if the number of tunable parameters in the global adaptation is reduced. We can choose appropriate settings in the global adaptation according to our needs.

Table 7: The results of different bottleneck ratios of the adapters.

| ratio $r$ | Pitts30k-test | | | MSLS-val | | |
|---|---|---|---|---|---|---|
| | R@1 | R@5 | R@10 | R@1 | R@5 | R@10 |
| $r$=0.25 | 91.0 | 96.2 | 97.4 | **91.2** | **96.6** | 96.9 |
| $r$=0.5 | **91.4** | **96.5** | **97.8** | 90.8 | 96.4 | **97.2** |
| $r$=0.75 | 90.9 | 95.9 | 97.2 | 91.1 | 96.4 | 96.6 |

Table 8: Performance of different global features without or with re-ranking.

| Method | Pitts30k-test | | | MSLS-val | | |
|---|---|---|---|---|---|---|
| | R@1 | R@5 | R@10 | R@1 | R@5 | R@10 |
| SelaVPR-cls(global) | 89.2 | 96.2 | 97.6 | 89.3 | 96.2 | **97.2** |
| SelaVPR-GeM(global) | 87.2 | 95.1 | 97.2 | 87.7 | 95.8 | 96.6 |
| SelaVPR-cls(re-rank) | 90.1 | 96.0 | 97.4 | 83.5 | 92.2 | 94.1 |
| SelaVPR-GeM(re-rank) | **91.4** | **96.5** | **97.8** | **90.8** | **96.4** | **97.2** |

Table 9: The results of using "local" features at different granularities for re-ranking.

| Method | Pitts30k-test | | | MSLS-val | | |
|---|---|---|---|---|---|---|
| | R@1 | R@5 | R@10 | R@1 | R@5 | R@10 |
| global adaptation for global retrieval | 87.3 | 94.6 | 96.6 | 87.4 | 95.9 | 96.9 |
| coarse patch tokens for re-ranking | 89.8 | 95.4 | 96.9 | 82.8 | 92.0 | 94.9 |
| dense local features for re-ranking | 91.4 | 96.5 | 97.8 | 90.8 | 96.4 | 97.2 |

## C  PERFORMANCE OF DIFFERENT GLOBAL FEATURES

This section compares the performance of using class token or GeM as global features in our SelaVPR method. As shown in Table 8, the performance of direct global retrieval (without re-ranking) using class token as the global feature, i.e. SelaVPR-cls(global), is better than that of GeM. However, SelaVPR-GeM(re-rank) is better than SelaVPR-cls(re-rank) after re-ranking using local features. That is, the improvement of SelaVPR-cls after re-ranking is significantly lower than that of SelaVPR-GeM, and the performance of SelaVPR-cls after re-ranking is even decreased on MSLS-val. This shows that when we use local features in conjunction with global features, GeM is better compatible with local features, whereas class token is not. Therefore we finally choose GeM as the global feature in SelaVPR.

## D  PERFORMANCE OF "LOCAL" FEATURES AT DIFFERENT GRANULARITIES

To illustrate the necessity of using dense local features in reranking, we provide a comparison of reranking using coarse 16×16 patch tokens (directly produced by the global adapted backbone, and treated as coarse-grained "local" features) and our dense local features. The results are shown in Table 9. Although re-ranking with coarse patch features provides an improvement (but not as good as ours) on Pitts30k compared to only global adaptation for global retrieval (without re-ranking), it performs significantly worse than global retrieval on MSLS-val, where VPR methods are more susceptible to perceptual aliasing. That is, it will bring negative effects and damage the results of global retrieval. However, using dense local features (produced by local adaptation) for re-ranking can always provide significant improvements compared to global retrieval.

## E  PERFORMANCE OF RE-RANKING DIFFERENT NUMBERS OF CANDIDATES

We show the results of our SelaVPR for re-ranking different numbers of candidates in Table 10. The performance roughly reaches saturation when re-ranking top-100 candidates, which is our recommended setting for optimal recognition performance. Our approach also achieves excellent performance when only the top-20 candidates are re-ranked. This setting can reduce the re-ranking runtime by approximately 80%.

## F  COMPARISONS ON MORE DATASETS

This section provides comparisons with other methods on more datasets. We compare the proposed method with other two-stage methods on the Nordland and St. Lucia datasets. As shown in Table 11,

Table 10: The results of re-ranking different numbers of candidates. Note that the model tested on Tokyo24/7 and Pitts30k-test is fine-tuned on MSLS-train and then further fine-tuned on Pitts30k-train (same as compared with the SOTA methods, but different from the ablation experiments).

| Candidates | Tokyo24/7 | | | Pitts30k-test | | | MSLS-val | | |
|---|---|---|---|---|---|---|---|---|---|
| | R@1 | R@5 | R@10 | R@1 | R@5 | R@10 | R@1 | R@5 | R@10 |
| Top-20 | 93.0 | 96.2 | 97.1 | 92.7 | 96.6 | 97.5 | 90.7 | 96.4 | 97.0 |
| Top-50 | 93.7 | 96.5 | 96.8 | 92.8 | 96.8 | 97.6 | 90.8 | 96.4 | 97.0 |
| Top-100 | 94.0 | 96.8 | 97.5 | 92.8 | 96.8 | 97.7 | 90.8 | 96.4 | 97.2 |
| Top-200 | 94.0 | 96.8 | 97.5 | 92.8 | 96.9 | 97.8 | 90.7 | 96.5 | 97.3 |

Table 11: Comparison to two-stage methods on Nordland-test and St. Lucia. The threshold is set to $\pm 2$ frames for Nordland-test and 25 meters for St. Lucia. The model is the same as that used for MSLS in Table 2.

| Method | Nordland-test | | | St. Lucia | | |
|---|---|---|---|---|---|---|
| | R@1 | R@5 | R@10 | R@1 | R@5 | R@10 |
| SP-SuperGlue | 25.8 | 35.4 | 38.2 | 86.5 | 92.1 | 93.4 |
| Patch-NetVLAD-s | 44.2 | 57.5 | 62.7 | 90.2 | 93.6 | 95.0 |
| Patch-NetVLAD-p | 51.6 | 60.1 | 62.8 | 93.9 | 95.5 | 96.2 |
| TransVPR | 61.3 | 71.7 | 75.6 | 98.7 | 99.0 | 99.2 |
| SelaVPR(global) | 72.3 | 89.4 | 94.4 | 99.4 | 99.9 | 100.0 |
| SelaVPR | **85.2** | **95.5** | **98.5** | **99.8** | **100.0** | **100.0** |

our method surpasses other methods even without re-ranking, i.e., SelaVPR(global). The SelaVPR outperforms other methods by a wide margin and achieves near-perfect results on St. Lucia (99.8% R@1 and 100% R@5). We also provide the results of our SelaVPR on Pitts250k in Table 12. MixVPR achieves the previous best results on Pitts250k, and SelaVPR outperforms it.

## G  QUALITATIVE RESULTS OF LOCAL MATCHING

We further illustrate that our method makes the produced features by foundation models more suitable for local matching in this section, and show the qualitative local matching results of our SelaVPR method and the pre-trained DINOv2 in Fig. 6. To make a fair comparison, the feature maps output by the ViT backbone of the two models are used for local matching. Our SelaVPR gets more correct matches (i.e. matching point pairs) than the pre-trained DINOv2 between two images from the same place. For two images from different places, all local matches are wrong and we expect as few matches as possible. SelaVPR produces fewer wrong matches than pre-trained DINOv2.

Table 12: The results of MixVPR and our method on Pitts250k. The model is the same as that used for Pitts30k in Table 2.

| Method | Pitts250k-test | | |
|---|---|---|---|
| | R@1 | R@5 | R@10 |
| MixVPR | 94.3 | 98.2 | 98.9 |
| SelaVPR | **95.7** | **98.8** | **99.2** |

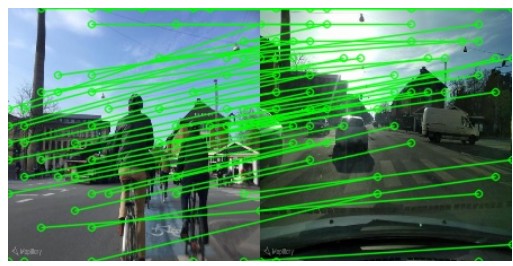 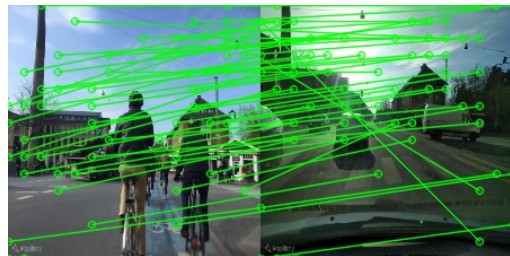

Produced by our SelaVPR (60 matches)    Produced by DINOv2 (57 matches)

(a) Local matching between two images from the **same place**

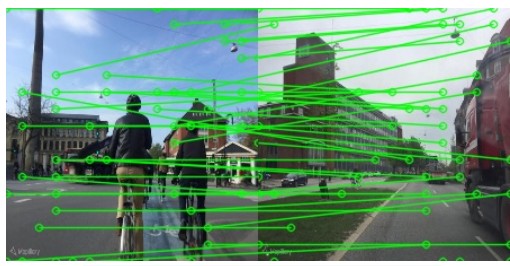 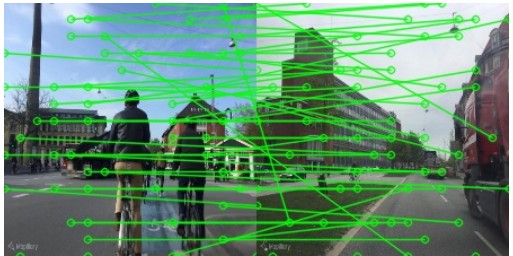

Produced by our SelaVPR (47 matches)    Produced by DINOv2 (55 matches)

(b) Local matching between two images from **different places**

Figure 6: Comparison of local matching between our SelaVPR and pre-trained DINOv2. (a) shows the local matching between images from the same place. The more matches the better, and the matches produced by ours are more than those of DINOv2. Note that some matches produced by DINOv2 are clearly inappropriate. (b) shows the local matching between images from different places. The fewer (wrong) matches the better, and our method has fewer (wrong) matches than pre-trained DINOv2.

## H    ADDITIONAL ATTENTION VISUALIZATION

Fig. 1 in the main paper has shown that the feature representation of our method can precisely focus on image regions that are helpful for place recognition. Here, Fig. 7 demonstrates more challenging examples. Compared to the pre-trained model, which is disturbed by objects such as dynamic foreground, our method always pays attention to all discriminative regions (buildings and vegetation).

## I    ADDITIONAL QUALITATIVE RESULTS

Fig. 4 in the main paper has shown a small number of qualitative results. Here, Fig. 8, Fig. 9 and Fig. 10 show more qualitative results on Tokyo24/7, MSLS, and Pitts30k, respectively. These examples show challenging cases such as severe condition changes, viewpoint changes, dynamic interference, and only small regions of discriminative landmarks or almost no landmarks. Our method obtains correct results, while other methods produce incorrect results.

## J    DATASET DETAILS

**Tokyo24/7** (Torii et al., 2015). The Tokyo24/7 dataset includes 75984 database images and 315 query images captured from urban scenes. The query images are selected from 1125 images taken at 125 distinct places with 3 different viewpoints and at 3 different times of day. This dataset mainly exhibits viewpoint changes and drastic condition changes (day-night changes).

**Mapillary Street-Level Sequences (MSLS)** (Warburg et al., 2020). The MSLS dataset is a large-scale VPR dataset containing over 1.6 million images labeled with GPS coordinates and compass angles, captured from 30 cities in urban, suburban, and natural scenes over seven years. It covers various challenging visual changes due to illumination, weather, season, viewpoint, as well as

Input images        Results of DINOv2        Results of ours

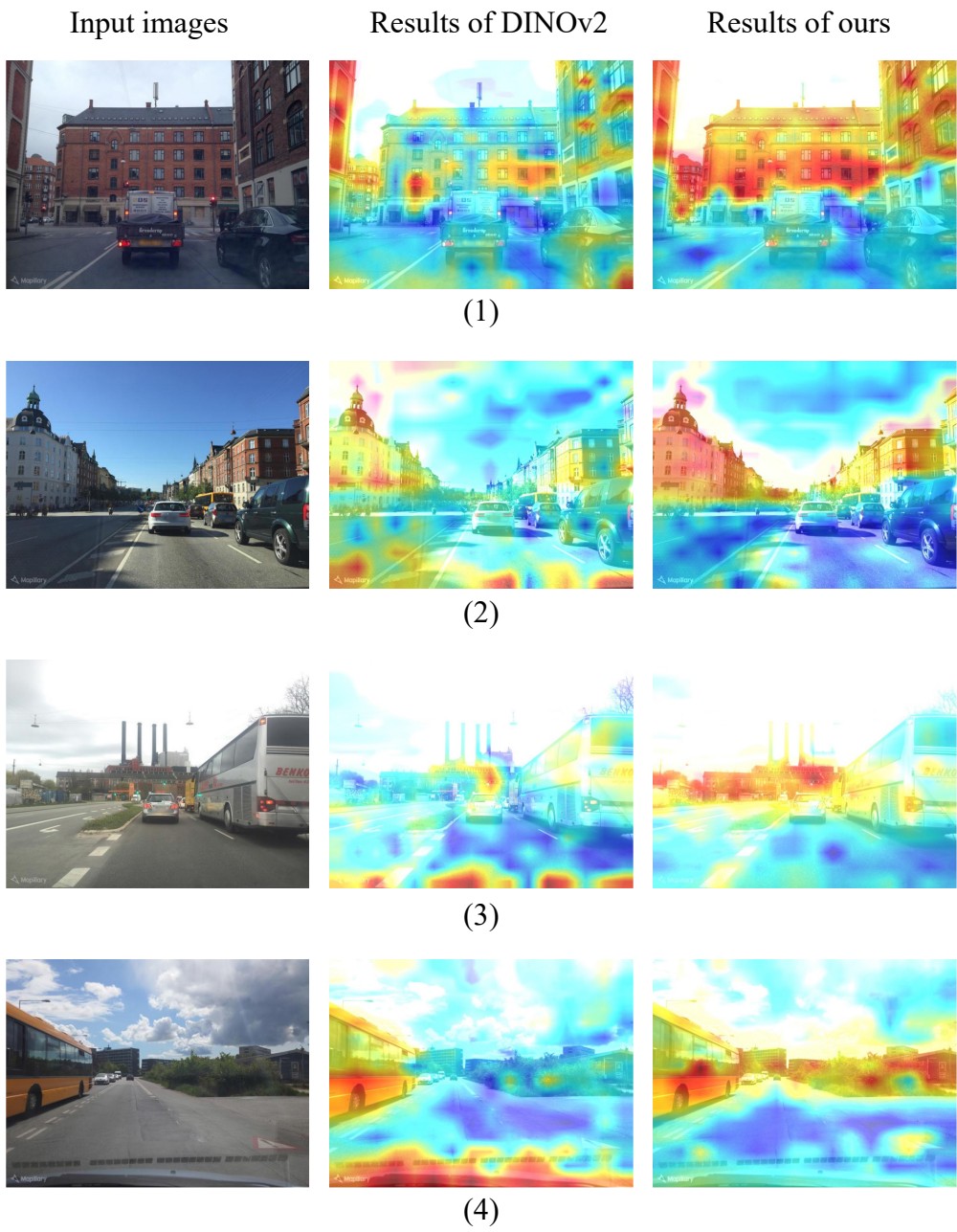

Figure 7: Additional attention map visualizations of the pre-trained model (DINOv2) and our model. We compute the mean in the channel dimension of the output feature map and display it using the heat map. In the last example, although our method is also affected by the bus, this is because it is closer and takes up a larger region of the image, while the buildings and vegetation are far away and occupy a smaller region. However, our method can still avoid missing any discriminative landmark (building and vegetation) to help retrieve correct results, while the pre-trained model cannot.

dynamic objects, and includes subsets of training, public validation (MSLS-val), and withheld test (MSLS-challenge). Following several related works (Hausler et al., 2021; Wang et al., 2022a; Zhu et al., 2023), the MSLS-val and MSLS-challenge sets are used to evaluate models.

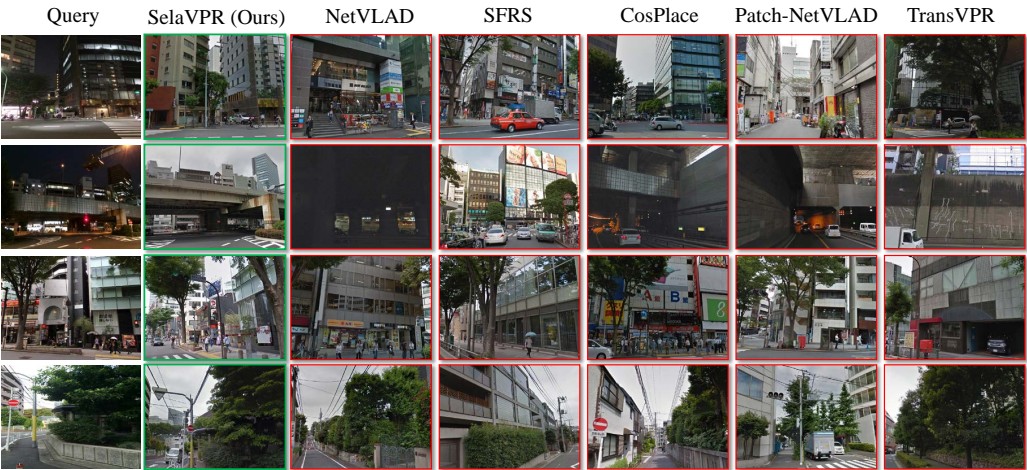

Figure 8: Qualitative results on Tokyo24/7. These challenging examples exhibit severe condition (lighting) changes and viewpoint changes.

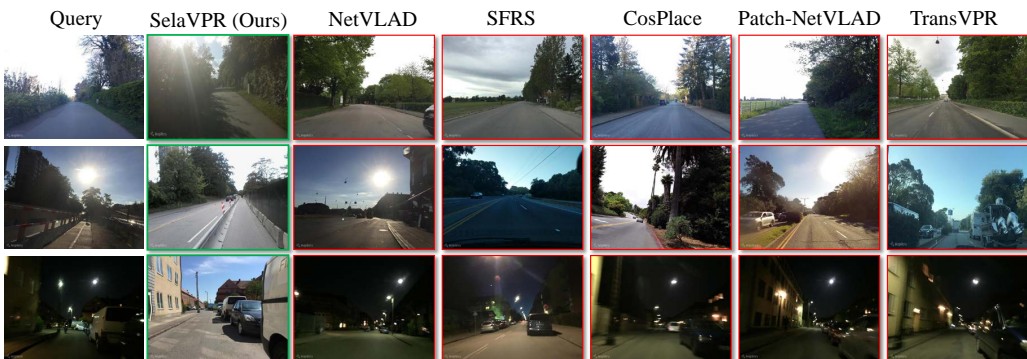

Figure 9: Qualitative results on MSLS. The first and second examples are dominated by vegetation (which tends to change over time). Most methods are prone to perceptual aliasing in these examples. The third example is extremely dark. All other methods return nighttime images but are wrong, only our method gets the correct image (but during daytime).

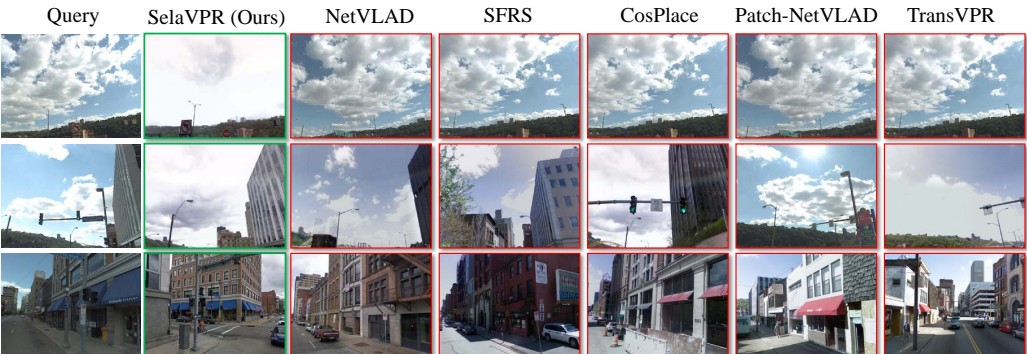

Figure 10: Qualitative results on Pitts30k. In the first and second examples, most of the image area is sky, and our method retrieves correct results using distinguishable landmarks that occupy only a small portion of the image region. The third example shows large viewpoint changes. Our approach is robust to these challenges.

**Pittsburgh** (Torii et al., 2013). The Pittsburgh dataset is collected from Google Street View panoramas, and provides 24 images with different viewpoints at each place. The images in this dataset exhibit large viewpoint variations and moderate condition variations. The Pitts30k dataset is a subset of Pitts250k (but more difficult than Pitts250k) and contains 10k database images each in the training, validation, and test sets.

**Nordland** (Olid et al., 2018). The Nordland dataset primarily consists of suburban and natural place images, captured from the same viewpoint in the front of a train across four seasons, which allows the images to show severe condition (e.g., season and light) changes but no viewpoint variations. Its ground truth is provided by the frame-level correspondence. Following the previous works (Olid et al., 2018; Wang et al., 2022a), we use the dataset partition first presented in (Olid et al., 2018) for our experiments. The summer (reference) and winter (query) images of the down-sampled version (224×224) of the test set (3450 images per sequence) are adopted to evaluate models.

**St. Lucia** (Glover et al., 2010; Berton et al., 2022b). The St. Lucia dataset comprises ten video sequences captured from the same suburban roadway in Brisbane. Following visual geo-localization benchmark (Berton et al., 2022b), the first and last sequences are used as the reference and query data in our experiments, and only one image is selected every 5 meters. Thus, the reference and query sequence include 1549 and 1464 images, respectively.

# K DETAILS OF THE COMPARED METHODS

**NetVLAD** (Arandjelovic et al., 2016). NetVLAD is a classic one-stage VPR method with a differentiable VLAD layer that can be integrated into neural networks. We use the pytorch implementation[1] with the released VGG16 model trained on the Pitts30k dataset.

**SFRS** (Ge et al., 2020)[2]. To train a more robust NetVLAD-based model, this work utilizes self-supervised image-to-region similarities to mine hard positive samples. The official model trained on Pitts30k is used in our experiments.

**CosPlace** (Berton et al., 2022a)[3]. This work introduces an extra large-scale dataset SF-XL and applies the training technique proposed in the classification task to train the VPR models. The official VGG16 model is used for comparisons.

**MixVPR** (Ali-Bey et al., 2023)[4]. This work presents a novel holistic feature aggregation method that takes feature maps from pre-trained backbones as global features, and uses a stack of Feature-Mixer to iteratively incorporate global relationships into each individual feature map. MixVPR is a SOTA one-stage VPR method. We use the best configuration (ResNet50 with 4096-dim output features) for comparisons.

**SP-SuperGlue** (DeTone et al., 2018; Sarlin et al., 2020)[5]. Hausler et al. (2021) first used this pipeline for the VPR task in the PatchNetVLAD work. SP-SuperGlue first retrieves candidate images using NetVLAD. Then, the SuperGlue (Sarlin et al., 2020) (a feature matcher based on graph neural network) is used to match SuperPoint (DeTone et al., 2018) features for re-ranking. The official implementation with the model trained on MegaDepth (Li & Snavely, 2018) is adopted.

**Patch-NetVLAD** (Hausler et al., 2021)[6]. This approach also retrieves candidates with NetVLAD, then re-ranks the candidates using the NetVLAD-based multi-scale patch-level features. We use the speed-focused and performance-focused configurations for evaluation. The official model trained on Pitts30k-train is tested on Pitts30k-test and Tokyo24/7, and the model trained on MSLS is tested on other datasets.

**TransVPR** (Wang et al., 2022a)[7]. This work first achieves candidate retrieval using the global features produced by integrating multi-level attentions from vision transformer, then utilizes an at-

---

[1] https://github.com/Nanne/pytorch-NetVlad
[2] https://github.com/yxgeee/OpenIBL
[3] https://github.com/gmberton/CosPlace
[4] https://github.com/amaralibey/MixVPR
[5] https://github.com/magicleap/SuperGluePretrainedNetwork
[6] https://github.com/QVPR/Patch-NetVLAD
[7] https://github.com/RuotongWANG/TransVPR-model-implementation

| # | User | Entries | Date of Last Entry | recall@5 ▲ |
|---|------|---------|---------------------|-----------|
| | | | Results | |
| 1 | anonymous02 | 1 | 09/17/23 | 0.89 (1) |
| 2 | | 1 | 03/14/23 | 0.88 (2) |
| 3 | | 5 | 06/25/23 | 0.84 (3) |
| 4 | | 11 | 07/27/22 | 0.82 (4) |
| 5 | | 9 | 05/18/23 | 0.80 (5) |
| 6 | | 2 | 04/21/22 | 0.80 (6) |
| 7 | | 1 | 09/26/23 | 0.80 (7) |
| 8 | | 10 | 03/01/22 | 0.77 (8) |
| 9 | | 3 | 10/16/22 | 0.77 (9) |
| 10 | | 9 | 10/16/22 | 0.76 (10) |
| 11 | | 3 | 10/10/22 | 0.74 (11) |
| 12 | | 20 | 10/22/22 | 0.74 (12) |
| 13 | | 5 | 04/04/23 | 0.74 (13) |
| 14 | | 9 | 05/23/23 | 0.73 (14) |
| 15 | | 4 | 02/20/22 | 0.71 (15) |

Figure 11: The snapshot of MSLS place recognition challenge leaderboard. Our SelaVPR method named "anonymous02" (for double-blind policy) ranks 1st at the time of submission.

tention mask to filter feature maps to get key-patch descriptors for re-ranking. The official model trained on Pitts30k-train is evaluated on Pitts30k-test and Tokyo24/7, while that trained on MSLS is tested on others.

**StructVPR** (Shen et al., 2023). To improve feature stability in a changing environment, StructVPR utilizes the segmentation images to enhance structural knowledge in global features and applies knowledge distillation to avoid online segmentation in testing. This method is combined with SP-SuperGlue to form a two-stage VPR method, and achieves great performance. Since the code is not released, we use the results reported in the original paper.

$R^2$**Former** (Zhu et al., 2023)[8]. This work addresses both the retrieval and re-ranking in two-stage VPR using a novel transformer model. Its re-ranking module takes feature correlation, attention value, and coordinates into account, and does not require RANSAC-based geometric verification. $R^2$Former is the SOTA two-stage VPR method. It ranked first in the MSLS challenge leaderboard before our method was proposed.

## L    THE SNAPSHOT OF MSLS LEADERBOARD

MSLS place recognition challenge [9] (Hausler et al., 2021; Zhu et al., 2023) is an authoritative competition for VPR with over 100 participants. Fig. 11 shows the snapshot of the MSLS challenge leaderboard at the time of submission. The proposed SelaVPR method (named "anonymous02" for double-blind policy) ranks 1st.

---

[8]https://github.com/bytedance/R2Former
[9]https://codalab.lisn.upsaclay.fr/competitions/865

