# OpenReview forum: "Towards Seamless Adaptation of Pre-trained Models for Visual Place Recognition"
_ICLR.cc/2024/Conference — ICLR 2024 poster_

### Official Review · Reviewer_Gibf · 2023-10-29

**Soundness:** 3 good
**Presentation:** 4 excellent
**Contribution:** 3 good
**Rating:** 8
**Confidence:** 4

**Summary:**

In this paper the authors propose a global-local adaptation method to seamlessly adapt the pre-trained DINOv2 model to produce both global and local features for the visual place recognition task. The proposed feature representation can focus on discriminative landmarks and eliminate dynamic interference. The output local features are used in local matching for re-ranking to further boost performance. This method outperforms other state-of-the-art methods on multiple datasets with high computational efficiency.

**Strengths:**

1.	This paper is very well written and clearly presented.
2.	Recapitulation of related work is good.
3.	The authors design a hybrid adaptation method to seamlessly adapt pre-trained foundation model to the VPR task. The method is novel and interesting, and the authors did not over-complicate it.
4.	The experimental results are really good.

**Weaknesses:**

1.	Pitts250k is also a common VPR dataset. The proposed approach has shown excellent results on multiple benchmark datasets. Providing the results on the Pitts250k dataset might make the experiment more complete.
2.	Re-ranking top-100 candidates seems a common setting for two-stage VPR methods. However some works also show the performance with different numbers of re-ranking candidates [1], which can help other researchers choose the optimal number of candidates when using this method. I think it is also necessary to show the performance of different numbers of candidates.

[1] Zhu, Sijie, et al. "R2former: Unified retrieval and reranking transformer for place recognition." Proceedings of the IEEE/CVF Conference on Computer Vision and Pattern Recognition. 2023.

**Questions:**

Please see the weaknesses.

---

> ### Author Response · Authors · 2023-11-20
> **Response to Reviewer Gibf**
>
> Thanks for your encouraging words and constructive comments.
>
> The results of our SelaVPR on Pitts250k are shown in the table below. MixVPR achieves the previous best results on Pitts250k, and SelaVPR outperforms MixVPR.
>
> |  Method  |   | Pitts250k  |  |
> | :---: | :---: | :---: | :---: |
> |   | R@1 | R@5 | R@10 |
> | MixVPR | 94.1 | 97.9 | 98.7 |
> | SelaVPR | 95.7 | 98.8 | 99.2 |
>
> The results of re-ranking different numbers of candidates are shown in the table below. Performance roughly reaches saturation when re-ranking top-100 candidates, which is our recommended setting for optimal recognition performance. Re-ranking top-50 (or even top-20) candidates is also a good option for faster two-stage retrieval.
>
> |  Method  |   | Tokyo24/7  |  |  |   | Pitts30k  |  |  |   | MSLS-val  |  |
> | :---: | :---: | :---: | :---: | :---: | :---: | :---: | :---: | :---: | :---: | :---: | :---: |
> |   | R@1 | R@5 | R@10 |  | R@1 | R@5 | R@10 |  | R@1 | R@5 | R@10 |
> | re-ranking top-20  | 93.0 | 96.2 | 97.1 |  | 92.7 | 96.6 | 97.5 |  | 90.7 | 96.4 | 97.0 |
> | re-ranking top-50  | 93.7 | 96.5 | 96.8 |  | 92.8 | 96.8 | 97.6 |  | 90.8 | 96.4 | 97.0 |
> | re-ranking top-100  | 94.0 | 96.8 | 97.5 |  | 92.8 | 96.8 | 97.7 |  | 90.8 | 96.4 | 97.2 |
> | re-ranking top-200  | 94.0 | 96.8 | 97.5 |  | 92.8 | 96.9 | 97.8 |  | 90.7 | 96.5 | 97.3 |

---

### Official Review · Reviewer_mSaw · 2023-10-30

**Soundness:** 4 excellent
**Presentation:** 3 good
**Contribution:** 3 good
**Rating:** 8
**Confidence:** 5

**Summary:**

The paper presents a hybrid global and local adaptation method to adapt pre-trained foundation models to two-stage visual place recognition. The global adaptation is achieved by adding parallel and serial adapters in each ViT block. The local adaptation is implemented by adding up-sampling layers after ViT backbone to produce dense local features. A novel mutual nearest neighbor local feature loss is proposed to train the local adaptation module. This architecture achieves fast two-stage place retrieval and outperforms several SOTA methods. It is ranked first on the MSLS challenge leaderboard.

**Strengths:**

The paper is well-organized and presents a good overview of the related work.
The approach is simple and easy to follow.
The experimental datasets are sufficient (conducted on 6 VPR benchmark datasets) and the results are excellent (outperform previous SOTA methods by a large margin).
This method can bridge the gap between the tasks of model pre-training and VPR using only a small amount of training data and training time. The two-stage retrieval runtime on Pitts30k is less than 0.1s (about 3% of the TransVPR method). This makes contributions to use pre-trained foundation models for real-world large-scale VPR applications.

**Weaknesses:**

This method achieves significantly better performance than other methods on several VPR datasets, and the authors qualitatively demonstrate some challenging examples. However, the motivation of the proposed method is not demonstrated well.  In particular, the gap of the tasks of model pre-training and VPR is not very clear to me. In addition, this paper does not show failure cases, which can inform future research in VPR.

**Questions:**

1. Will re-ranking more candidate places achieve better performance or hurt the results?
2. This work finetunes the models on the MSLS dataset and further finetunes them on Pitts30k to test on Pitts30k and Tokyo24/7, which is the same as R2Former. However, the R2Former work provides the result on Pitts30k of the model that only trained on MSLS, which can prove the model's transferability to the domain gap. Can the proposed SelaVPR still outperform R2Former on Pitts30k using only MSLS for training?

---

> ### Author Response · Authors · 2023-11-20
> **Response to Reviewer mSaw**
>
> Thanks for your positive recommendation and insightful comments.
>
> **[Q1:]** Will re-ranking more candidate places achieve better performance or hurt the results?
>
> **[A1:]** Re-ranking more candidates neither improves nor hurts performance. The results of re-ranking different numbers of candidates are shown in the table below. Performance roughly reaches saturation when re-ranking top-100 candidates, which is our recommended setting for optimal recognition performance.
>
> |  Method  |   | Tokyo24/7  |  |  |   | Pitts30k  |  |  |   | MSLS-val  |  |
> | :---: | :---: | :---: | :---: | :---: | :---: | :---: | :---: | :---: | :---: | :---: | :---: |
> |    | R@1 | R@5 | R@10 |  | R@1 | R@5 | R@10 |  | R@1 | R@5 | R@10 |
> | re-ranking top-20  | 93.0 | 96.2 | 97.1 |  | 92.7 | 96.6 | 97.5 |  | 90.7 | 96.4 | 97.0 |
> | re-ranking top-50  | 93.7 | 96.5 | 96.8 |  | 92.8 | 96.8 | 97.6 |  | 90.8 | 96.4 | 97.0 |
> | re-ranking top-100  | 94.0 | 96.8 | 97.5 |  | 92.8 | 96.8 | 97.7 |  | 90.8 | 96.4 | 97.2 |
> | re-ranking top-200  | 94.0 | 96.8 | 97.5 |  | 92.8 | 96.9 | 97.8 |  | 90.7 | 96.5 | 97.3 |
>
>
> **[Q2:]** Can the proposed SelaVPR still outperform R$^2$Former on Pitts30k using only MSLS for training?
>
> **[A2:]** The results of R$^2$Former and our SelaVPR on Pitts30k using only MSLS for training are shown in the table below. Our SelaVPR also outperforms R$^2$Former. The results of SelaVPR trained only on MSLS are even better than those of SelaVPR only trained on Pitts30k (similar to the case with R$^2$Former), showing that our method has good transferability as well.
>
> |  Method  |   | Pitts30k  |  |
> | :---: | :---: | :---: | :---: |
> |    | R@1 | R@5 | R@10 |
> | R$^2$Former | 88.4 | 94.2 | 95.7 |
> | SelaVPR | 92.2 | 96.5 | 97.6 |
>
> For weaknesses: The gap between the model pre-training and VPR tasks primarily refers to the difference in training objectives and data, which further results in the pre-trained models focusing on different objects from the specially trained VPR models. A robust VPR model should focus on the static discriminative backgrounds (e.g., buildings and trees) and ignore the dynamic foreground objects (e.g., pedestrians and vehicles), but a pre-trained model cannot do this, as vividly shown in the attention map visualizations in our paper.
> The failure cases mainly occur in some natural scenes that lack discriminative landmarks, which is difficult to address for existing VPR methods. It’s the focus of our future work.

---

### Official Review · Reviewer_S5xw · 2023-10-31

**Soundness:** 4 excellent
**Presentation:** 3 good
**Contribution:** 4 excellent
**Rating:** 8
**Confidence:** 5

**Summary:**

1.	Visual place recognition (VPR) is a fundamental task for applications in robot localization and augmented reality. This paper aims to bridge the gap between the tasks of pre-training and VPR, thus fully unleashing the capability of pre-trained models for VPR.
2.	The authors introduce a hybrid adaptation method to get both global features for retrieving candidate places and dense local features for re-ranking.
3.	Meanwhile, a novel local feature loss is designed to guide the production of proper local features for local matching without geometric verification in re-ranking.

**Strengths:**

1.	This work is novel and solid. The proposed SelaVPR is a well-designed method and achieves very fast two-stage retrieval.
2.	This paper makes two technically strong contributions: closing the gap between the pre-training and VPR tasks, and outputting proper dense local features for VPR task using DINO v2. The extensive ablation experiments and visualization results show that the proposed method well adapts the pre-trained model to the VPR task. The produced dense local features also perform well in local matching re-ranking.
3.	This method achieves better performance than the SOTA methods with less training data and training time.

**Weaknesses:**

1.	This work adapts the pre-trained model to the VPR task. The global and local features produced by this hybrid adaptation seem to be useful for more visual tasks. Expanding the use of this method can make the contribution of this paper more obvious.
2.	The clarity of the paper could be further improved.

**Questions:**

1.	Why is L2 distance used to measure global feature similarity, but dot product used to calculate local feature similarity?
2.	Is it feasible to re-rank top-k candidate images directly using coarse patch tokens from ViT? and how is the performance?

---

> ### Author Response · Authors · 2023-11-20
> **Response to Reviewer S5xw**
>
> Thanks for your positive recommendation and valuable suggestions.
>
> **[Q1:]** Why is L2 distance used to measure global feature similarity, but dot product used to calculate local feature similarity?
>
> **[A1:]** To make the first stage (global retrieval) compatible with some one-stage VPR methods and the benchmark [1], the L2 distance is used to compute global feature similarity. For the convenience of finding mutual matches (i.e. fast computing), local feature similarity is based on the inner product. Since local features have been L2 normalized, the inner product is equal to cosine similarity.
>
> **[Q2:]** Is it feasible to re-rank top-k candidate images directly using coarse patch tokens from ViT? and how is the performance?
>
> **[A2:]** It’s feasible in terms of program implementation. However, Re-ranking candidates with the coarse patch tokens from ViT does not make sense in two-stage VPR. Although it provides an improvement (but not as good as ours) on Pitts30k compared to global retrieval, it performs significantly worse than global retrieval on MSLS-val, where VPR methods are more susceptible to perceptual aliasing. That is, re-ranking with coarse patch features will bring negative effects and damage the results of global retrieval (i.e. the first stage retrieval).
>
> |  Method  |   | Pitts30k  |  |  |   | MSLS-val  |  |
> | :---: | :---: | :---: | :---: | :---: | :---: | :---: | :---: |
> |    | R@1 | R@5 | R@10 |  | R@1 | R@5 | R@10 |
> | global adaptation for global retrieval  | 87.3 | 94.6 | 96.6 |  | 87.4 | 95.9 | 96.9 |
> | coarse patch tokens for re-ranking  | 89.8 | 95.4 | 96.9 |  | 82.8 | 92.0 | 94.9 |
> | dense local features for re-ranking  | 91.4 | 96.5 | 97.8 |  | 90.8 | 96.4 | 97.2 |
>
> For the weaknesses, we will improve the clarity in the revised paper and try to use the hybrid adaptation method for other visual tasks in future work. Thanks again for your suggestions.
>
> Reference
>
> [1] Gabriele Berton, Riccardo Mereu, Gabriele Trivigno, Carlo Masone, Gabriela Csurka, Torsten Sattler, and Barbara Caputo. Deep visual geo-localization benchmark. In Proceedings of the IEEE/CVF Conference on Computer Vision and Pattern Recognition, pp. 5396–5407, 2022.

---

### Official Review · Reviewer_LtSJ · 2023-11-02

**Soundness:** 3 good
**Presentation:** 2 fair
**Contribution:** 2 fair
**Rating:** 5
**Confidence:** 5

**Summary:**

The authors propose a method to adapt foundation models to the task of visual place recognition, arguing that the object-centric focus of the training of foundation models does not align with the background/static-object attention needed in the VPR task. Thus, they propose, instead of fine-tuning the visual transformers, to extend the transform block with two mechanisms (added MLP) that operate as adapter for the global feature computation. Another local adaptation is done for reranking, similar to geometric verification of retrieved images.

**Strengths:**

+ concept is straightforward: the idea behind the adaptation, and the motivation why it is needed to adapt from foundation models is clearly presented.
+ experiments on relevant datasets: results are very good, improving on existing approaches
+ the paper is easy-to-read

**Weaknesses:**

_very limited or non-existing insights_
the paper is built solely around overcoming the results of existing methods, while insights and evidence-based contributions are not provided. After reading the paper I am left with a big question: "why this method works and what do I learn that can perhaps use to design methods in different applications?". Would the adaptation work also for CNNs pre-trained on ImageNet (as they share the same object-centric bias)?
I would expect (at ICLR) a thorough analysis of reasons why the performance are much higher, what the implications of doing adaptation are, and what are the real scientific contributions behind this work (not just that the method gets better results than sota methods).

_design choice weakly explained_
no motivations or justification of why the adapters are designed in a certain way, and what the difference w.r.t. existing approaches for adaptation of foundation models are. What is the hypothesis behind this kind of design, and what explanations can be given (with experimental evidence) about their working principle?

_data-efficiency not elaborated upon_
as data-efficiency is a key argument about using foundation models, the authors indeed mention it but do not provide substantiable experimental evidence about how it benefits their approach.

_parameter difference not well-analyzed_
The adaptation mechanisms proposed are still requiring the fine-tuning of +50M parameters, which is much more than other methods train. Summed up with the +300M parameters of the foundation backbones, these models account for much more capacity than whatever method used previously. The authors do not provide any discussion about this point, or experiments with adaptation of other (smaller) models.

A missing reference:
Leyva-Vallina et al., Data-Efficient Large Scale Place Recognition With Graded Similarity Supervision; CVPR 2023

**Questions:**

- How would the adapters work with other (smaller) models?
- What are the reasons, and interpretaions (with evidences) of why the proposed adapters work?
- How the adapter parameter space influences the improve of performance, and how does it relate with 'smaller' backbones?

---

> ### Author Response · Authors · 2023-11-20
> **Response to Reviewer LtSJ (1/2)**
>
> Thanks for your comments and suggestions. We hope the following clarifications help to address your concerns.
>
> **[Q1:]** How would the adapters work with other (smaller) models?
>
> **[A1:]** The adapters can work with smaller models in the way described in the paper and achieve good performance. But it only makes sense if this model is pre-trained on a large amount of data and produces well-generalized representations. Otherwise, we can directly fully fine-tune the model. Here we provide the results (only fine-tuning the model on Pitts30k and testing on Pitts30k) of our adaptation method using a distilled version of DINOv2 (based on ViT-Base) as backbone, denoted as DINOv2-B. The DINOv2-L (based on ViT-Large) is the backbone used in our paper. Their performances are close, demonstrating that our adaptation method can be used with smaller models.
>
> | Backbone  | R@1 | R@5 | R@10 |
> | :---: | :---: | :---: | :---: |
> | DINOv2-L   | 91.4 | 96.5 | 97.8 |
> | DINOv2-B   | 90.5 | 96.1 |97.1 |
>
> **[Q2:]**  What are the reasons why the proposed adapters work?
>
> **[A2:]**  Although visual foundation models can provide powerful feature representations, the objects that need to be focused on may be different for various downstream tasks. Freezing the foundation model can retain the powerful representation capabilities, while inserting the lightweight trainable adapters so that the entire model can be adapted to the VPR task after fine-tuning on VPR datasets, that is, the trained adapter has learned VPR-related knowledge. Our visualization results well illustrate that the adapted model can focus on objects that are helpful for place recognition and ignore dynamic foreground interference, which is the key reason for its good performance. Meanwhile, our hybrid adaptation method combines global adaptation and local adaptation, which not only yields a robust global representation, but also enables the model to output dense local features for spatial matching re-ranking, thereby further boosting performance.
>
> **[Q3:]** How the adapter parameter space influences the improve of performance, and how does it relate with 'smaller' backbones.
>
> **[A3:]** The number of parameters of the adapter in our global adaptation is adjustable. Appropriately increasing the parameters of the adapter can slightly improve performance. However, in most cases, we can actually use a very lightweight adapter to obtain good performance. In the Appendix.C of our paper, we have shown the results of using only one adapter in each transformer block or reducing the bottleneck ratio to 0.25, and the performance is still good. Here, we provide the results (on Pitts30k) of using only the parallel adapter and reducing the bottleneck ratio from 0.5 (in the paper) to 0.05, as shown in the table below (SelaVPR** uses the more lightweight adapter. SelaVPR represents the original setup in paper). The SelaVPR** has only 5.19M trainable parameters (less than most methods), and the results are still excellent. For smaller backbones, fewer transformer blocks and lower token dimensions will naturally lead to fewer parameters of the adapter. Overall, the original paper focused more on recognition performance without excessively limiting the number of parameters. In fact, our method can achieve good performance using a significantly smaller number of parameters.
>
> | Method  | Tunable Param (M)  | R@1 | R@5 | R@10 |
> | :--- | :---: | :---: | :---: | :---: |
> | ResNet50-GeM |  7  |  82.3 |  91.9 |  94.6 |
> | SelaVPR  |  53 |  91.4 |  96.5 |  97.8 |
> | SelaVPR** |  5 |  91.2 |  96.4 |  97.4 |

---

> ### Author Response · Authors · 2023-11-20
> **Response to Reviewer LtSJ (2/2)**
>
> The following are Responses for the Weaknesses:
>
> **1. About the insights and contributions**
>
> Insights: The pre-trained foundation models (e.g. DINOv2) have achieved remarkable performance on many vision tasks given their ability to produce well-generalized representations. The recent AnyLoc work [1] also uses DINOv2 as the backbone, but without any adaptation or fine-tuning. Although AnyLoc uses a larger version backbone (based on ViT-giant), its performance is obviously worse than ours. Our work shows that there is a gap between the tasks of model pre-training and VPR (as we described in the third paragraph of the Introduction section). Model adaptation is a necessary thing to fully unleash the capability of pre-trained foundation models for VPR.
>
> Contributions: The main contribution is the hybrid global-local adaptation method to bridge the gap and fully unleash the capability of pre-trained models for VPR. Our work provides an effective and efficient way to seamlessly adapt pre-trained foundation models to produce both global and (dense) local features for two-stage VPR. Moreover, we propose a mutual nearest neighbor local feature loss to train the local adaptation module, which is combined with global feature loss for fine-tuning. Different from other two-stage methods, the local features produced by our model are directly used in cross-matching for re-ranking, without time-consuming geometric verification.
>
> For different applications: Our method is designed to seamlessly adapt the pre-trained foundation model for the VPR task. However, we think the hybrid adaptation method (to get both global features and local features) may also be useful for other applications.
>
> Would it work for CNN: For CNN models pre-trained on ImageNet, adaptation is not necessary. We can fully fine-tune such a model directly. Most of the adaptation methods are designed for the foundation models (especially these transformer-based models) pre-trained on a large amount of data (larger than ImageNet).
>
> About the performance: The powerful representation ability of the foundation model is the basis for good performance. But more importantly, our method bridges the gap between the tasks of model pre-training and VPR through seamless adaptation, i.e., the implication of doing adaptation is bridging the gap. The adapted model can produce the global features and local features more suitable for VPR to address the challenges in VPR tasks and achieve excellent performance.
>
> **2. Design motivations**
>
> Please see A2 for Q2. The design of our global adaptation is inspired by previous adapter-based work. However, we design a simple local adaptation method to adapt the ViT-based foundation models to produce dense local features for the local cross-matching in two-stage VPR. Our hybrid adaptation method combines the global adaptation and local adaptation to produce both global features and dense local features for two-stage VPR, which is different from previous adaptation works.
>
> **3. Data efficiency**
>
> The adaptation method can keep the powerful pre-trained image representations intact. When there is only insufficient training data in the downstream task, it mitigates the risk of over-fitting. This is a benefit shared by most adaptation methods. It is also an advantage of our approach over most other VPR methods. The experimental evidence is in the Appendix.A of our paper.
>
> **4. Model parameters**
>
> Please see A1 & A3 for the Q1 & Q3.
>
> **5. A missing reference**
>
> Thanks for the reminder. We’ll mention this work in our revised paper.
>
> Reference
>
> [1] Nikhil Keetha, Avneesh Mishra, Jay Karhade, Krishna Murthy Jatavallabhula, Sebastian Scherer, Madhava Krishna, and Sourav Garg. Anyloc: Towards universal visual place recognition. arXiv preprint arXiv:2308.00688, 2023.

---

### Meta-Review · Area_Chair_3D29 · 2023-12-05

**Metareview:**

This work aims to bridge the gap between the tasks of pre-training and visual place recognition and presents a hybrid global-local adaptation method to adapt the pre-trained visual foundation model to produce both global and local features for the two-stage visual place recognition. The majority of reviewers support the acceptance of this paper. The concerns of Reviewer LtSJ are mainly about insights and scientific contributions of this work. Authors addressed these concerns in response and I encourage authors to include the reviewer suggestions in the final version of the paper.

**Justification For Why Not Higher Score:**

The clarity and some details of the paper can be further improved.

**Justification For Why Not Lower Score:**

Reviewers agree that the experimental results are very good and the motivation is clearly presented. This work could be helpful to further development in the community.

---

### Decision · Program_Chairs · 2024-01-16

Accept (poster)